# Spatial Indices for Convivial Greenstreets

Emmy Kriehn, Kenneth Tamminga *  and Travis Flohr

Department of Landscape Architecture, Penn State University, University Park, PA 16802, USA;
emmy.titcombe@gmail.com (E.K.); tlf159@psu.edu (T.F.)
* Correspondence: krt1@psu.edu

**Abstract:** Streetside gardening is an informal, resident-initiated activity undertaken in dense urban areas worldwide. Yardless urban areas with a high incidence of informal streetside gardening are called Convivial Greenstreets (CG). Site investigations in European and several U.S. cities over the last decade suggest that social, ecological, and local climate benefits may be found where CG are most intense. The aim of this research is to fill a gap in the research literature by better understanding the spatial distribution of CG and the potential benefits associated with them. Using inner-core neighborhoods in Delft, The Netherlands, and Philadelphia, USA, as test cases, we devised a Convivial Greenstreet Intensity (CGI) index to provide a consistent method for mapping and comparing levels of streetside gardening activity across neighborhoods and cities. We show that CG spatial patterning and quantification of informal gardening intensity using in situ documentation and integrated GIS and Google Earth analyses are feasible and should prove useful as a basis for further research. With the development of a reliable method for measuring and mapping informal streetside gardening activity with a focus on visually accessible biomass, we hope that opportunities for investigating links between convivial greenstreets and urban microclimatic and physical and mental health will be facilitated.

**Keywords:** convivial greenstreets; urban gardening; green infrastructure; landscape urbanism; urban design

## 1. Introduction

Gardening is a fundamental human activity that is carried out in rural areas and cities alike. In dense urban areas where most residents have no access to outdoor yards, people may find varied and creative ways of gardening within the streetscape corridor itself. When a high level of streetside gardening activity occurs in relatively yardless neighborhoods, it is called a convivial greenstreet [1]. Along convivial greenstreets (CG), plantings and associated paraphernalia are not specified by urban professionals. Instead, these installations are largely personal expressions that imply residents' grassroots engagement in everyday community-building and place-making via hands-on horticulture.

Our focus in this paper is the spatial patterning and quantification of convivial greenstreet activity. We take a holistic interest in this emerging and, to date, minimally studied urban phenomenon. Our scope considers both the physicality of greenstreets, their potential for enhanced ecosystem services in the city, and their context as settings for various forms of urban micro-socialization along the sidewalk. Additionally, we seek to test the efficacy of enhancing in situ streetside documentation with ArcGIS Pro v. 3.0.3 and Google Earth v. 9.0 software applications that remote researchers can share online.

While convivial greenstreets are a novel and as-yet poorly understood part of urban morphology, the theoretical basis for CG inquiry has several threads that reach to the mid-20th century and earlier. Insights are offered from the fields of urban sociology and anthropology, architectural critique, and applied urbanism. In his seminal work, Tools for Conviviality, Ivan Illich set the stage for such greenstreets: "People… need above all the freedom to make things among which they can live, to give shape to them according

to their own tastes, and to put them to use in caring for and about others" [2] (p. 11). Indeed, the roots of place-based conviviality in a civil society run deep; Alison Gruseke [3] traced the idea back to the Garden of Eden and its "web of conviviality" between Maker, body, soil, and garden. Similarly, Michael Given [4] (p. 140) examined the relationship between humans and soil, writing, "In a convivial landscape, we humans think with soil, and the soil thinks with fungal networks, microbial action, symbiotic relationships, ion exchange—and us." Jane Jacobs paralleled these theories on conviviality at the community scale in her focus on fine-grained cities, writing "The trust of a city street is formed over time from many, many little public sidewalk contacts" [5] (p. 56).

Several years later, the guerrilla gardening movement involving informal (and often illicit) gardening on urban vacant lots appeared in New York City as an outgrowth of the street-based anti-war movement of the mid-1960s and as a counterpoint to growing urban polarization, isolation, and civic lethargy [6]. Decades later, this pro-local, anti-establishment tendency of some urban civil societies remains alive and expressive on the street. For example, in the Netherlands, the Utrecht organization GuerrillaGardeners.nl serves as a national clearing house for advocacy and training that promotes convivial greenstreets [7].

Research on urban liminality explores transitionary city spaces and time periods. In the highly diverse context of Dutch cities, researchers found that recent immigrants and asylum seekers used multi-lingual and multi-cultural adaptation to negotiate and thus feel part of otherwise inhospitable urban spaces [8]. Similarly, in their study of so-called "third spaces" along a New England main (high) street, Mehta and Bosson [9] (p. 802) found that personalized street fronts "often added a sense of delight, and occasionally, a sense of humor that could be shared by all". Although they did not specifically list streetside gardening, Maununaho et al. [10] (p. 19) called for "urban oases" as repositories of conviviality in the context of socio-economic diversity. They asserted that "urban nature can be enjoyed alone, or the experience can be shared with friends or strangers in the same place", creating settings for diverse social engagement.

Urban design theory also contributes contextuality to the notion of socially robust greenstreets. Wise and Noble [11] (p. 427) noted that material environs are as important as interpersonal relations in establishing conviviality and that "the flow of bodies through public space . . . represent affordances of conviviality through shared social resources." And presaging convivial greenstreets, Malcolm Miles called for the integration of biological, social, and cultural needs in city neighborhoods, asserting that conviviality ". . .leads to a creativity which is localized and self-sustaining" [12] (p. 228).

Although most such gardening occurs in niches, it is anything but a niche activity along many European and North American streetscapes. In some cities, informal streetside gardening is increasingly holding its own alongside walking, biking, and window shopping as a primary activity of daily life on the street. The cultivation of plants as a lingua franca that facilitates connections between diverse actors on the street is seen during almost every street research event [13]. Other empirical research confirmed the role of urban public spaces, and streets in particular, as contexts for this intense and genuine form of sociality in the city. Hinchliffe and Whatmore documented conviviality as simply accommodating social differences in urban spaces [14]. Thrift described conviviality as the everyday, banal, affective, and relational aspects of city life, noting that such encounters are pre-cognitive, affective, and emotional [15]. Nowicki and Vertovec [16] (p. 348) cited Christov-Bakargiev and Rolnik in defining conviviality as the human capacity to relate to the world. They note that conviviality "invites all kinds of people to take time, get affected by the environment, and co-create the space and situation for togetherness to happen." Wood et al. found that, in contrast to brisk walking, leisurely walking in the city was associated with a "sense of community", or the creation of social capital [17]. We can see, then, that streetside plants and active gardening could easily be included in the kind of conviviality generated by such interpersonal engagements.

Overall, our work suggests that the microcosm of convivial greenstreets is enabled via the social "friction" of streetside horticulture: slowing down, tending plants, using and sharing one's senses, and finding commonality around plants, growing media, and related paraphernalia [13]. While conviviality is unlikely to resolve inter-cultural urban acrimony at larger scales, it offers some counterpoint to the isolation and separateness that can result from rigid and coarse-grained cosmopolitanism. As Jacobs observed six decades ago, "Most (sidewalk interaction) is ostensibly utterly trivial, but the sum is not trivial at all" [5] (p. 56).

If self-expression via gardening is an important activity that enhances both individual and shared well-being in the city, then it will be increasingly important for policymakers to understand how to support these informal gardening activities without over-regulating or prescribing. Our research also suggests that ecological and local-scale climate benefits may be found where CG are most intense. Other researchers have provided further scientific context. For example, Simao et al. [18] showed that small flower plantings, such as those found on convivial greenstreets, have positive effects on small bee communities in urban systems. Additionally, Theodorou et al. [19] suggested urban areas can serve as hotspots of pollinator abundance and biodiversity. The order Hymenoptera and the *Bombus* species are particularly notable for the richness and variability of their foraging and nesting activities in areas of high edge density. Moreover, the robust vegetation of convivial greenstreets can bolster the evaporative cooling provided by the urban forest managed by civic officials [20,21]. Thus, knowing the location and characteristics of convivial greenstreets is necessary to understand their contributions to urban landscape patterns, connectivity, biodiversity, and thermal comfort. As both a social phenomenon and a distinctive biophysical component of urban morphology, convivial greenstreets are poised for closer scrutiny.

To shed further light on this emerging phenomenon, we needed to understand where convivial greenstreets already exist and analyze their spatial characteristics. This is most effectively carried out in terms of the intensity of convivial greenstreets activity and its aggregated distribution in a definable urban neighborhood or precinct. The convivial greenstreets intensity index (CGI Index) introduced below provides a novel methodological base for comparative convivial greenstreets research on several fronts. This includes the authors' ongoing investigation of possible links between microclimatic amelioration and greenstreets in the context of urban heat island buildup in the Netherlands and Pennsylvania [22]. We hope others see the utility of this method as green infrastructure research on the relationships between convivial greenstreets, urban sustainability, and livable civil society gain traction.

## 2. Materials and Methods

The methodology was divided into three steps. The first step consisted of creating the CGI Index. The second step consisted of rating street segments within each study area. The third step involved assessing the interrater reliability of the CGI Index. A general outline of the methods is provided in Figure 1 and the details of each step and requisite materials are elaborated below.

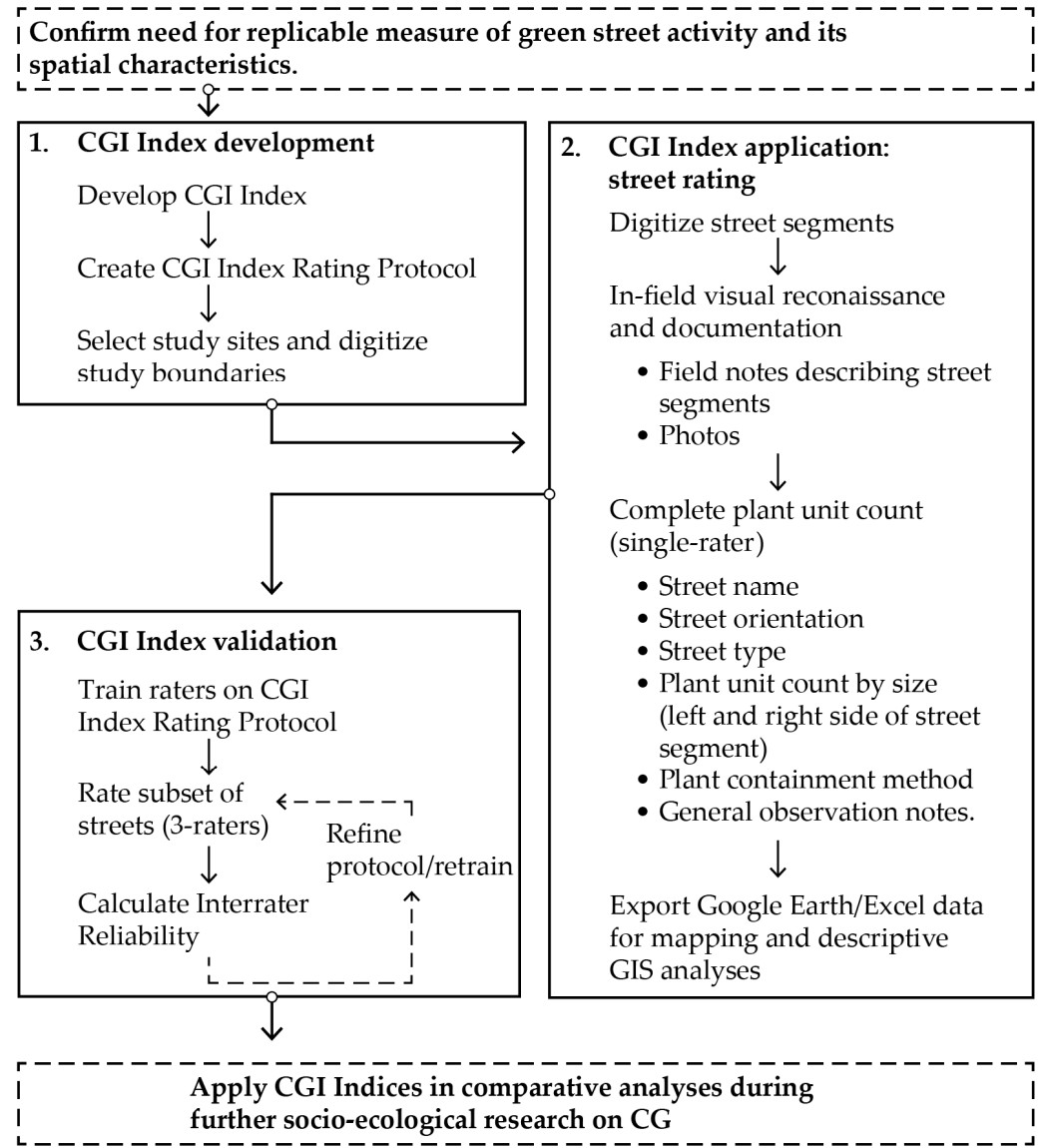

**Figure 1.** Research methodology diagram highlighting critical steps. Source: own work.

*2.1. Study Areas*

The CGI Index described herein focuses on two cities: Delft, The Netherlands, and Philadelphia, USA. These study sites were chosen because both displayed ample signs of greenstreet activity in their respective regions. Our reconnaissance was initially conducted between 2012 and 2020 in 31 European cities involving just over 80 neighborhoods. These initial forays indicated that the urban core of Delft (Figure 2) had some of the most robust and consistent greenstreet activity in the Netherlands and, indeed, compared with the +30 western Europe cities investigated. As a proof-of-concept exercise, Delft's well-defined and dense core was easier to delineate than several comparably verdant but less well-defined neighborhoods in Amsterdam, such as Jordaan and Weesperzijde. Delft's well-defined greenstreet geography accommodated informal gardening along almost every residential street. Next, we expanded from our broad base of European cities to include Philadelphia to begin comparative CG analyses between European and North American cities. Philadelphia was chosen because it was easily accessible to the authors and because we were familiar with emerging CG activity in several neighborhoods based on prior professional activities in the city. The inner-ring neighborhoods of Fishtown and Fitler

Square in Philadelphia (Figures 3–5), in particular, were comparable to Delft in terms of informal greenstreet activity.

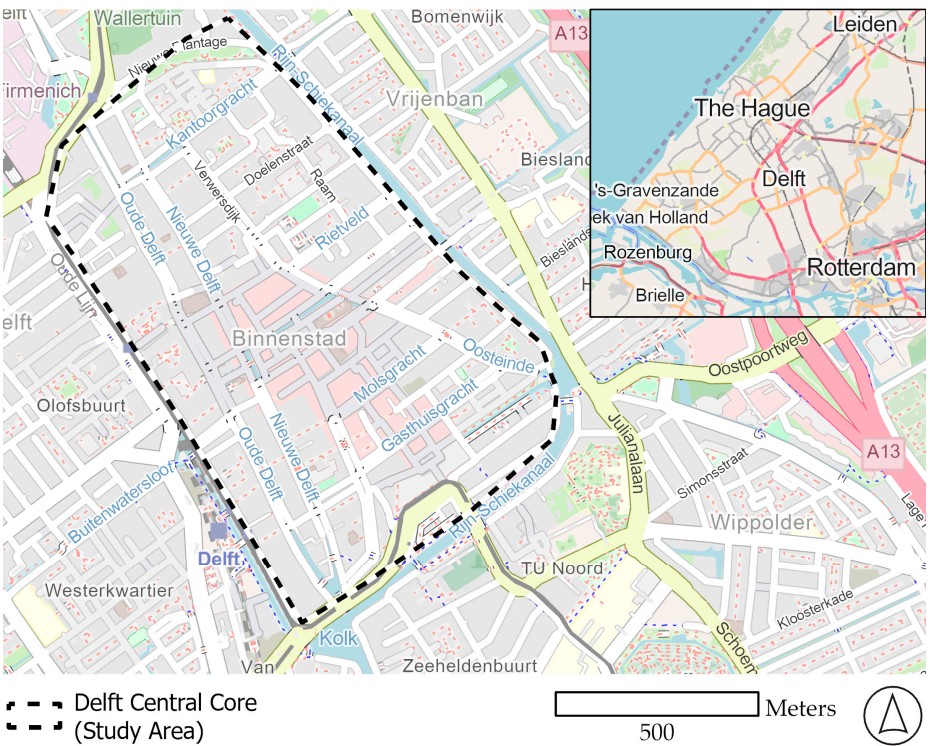

**Figure 2.** Study area in the central core of Delft, The Netherlands. Source: author created map using Esri's base map background.

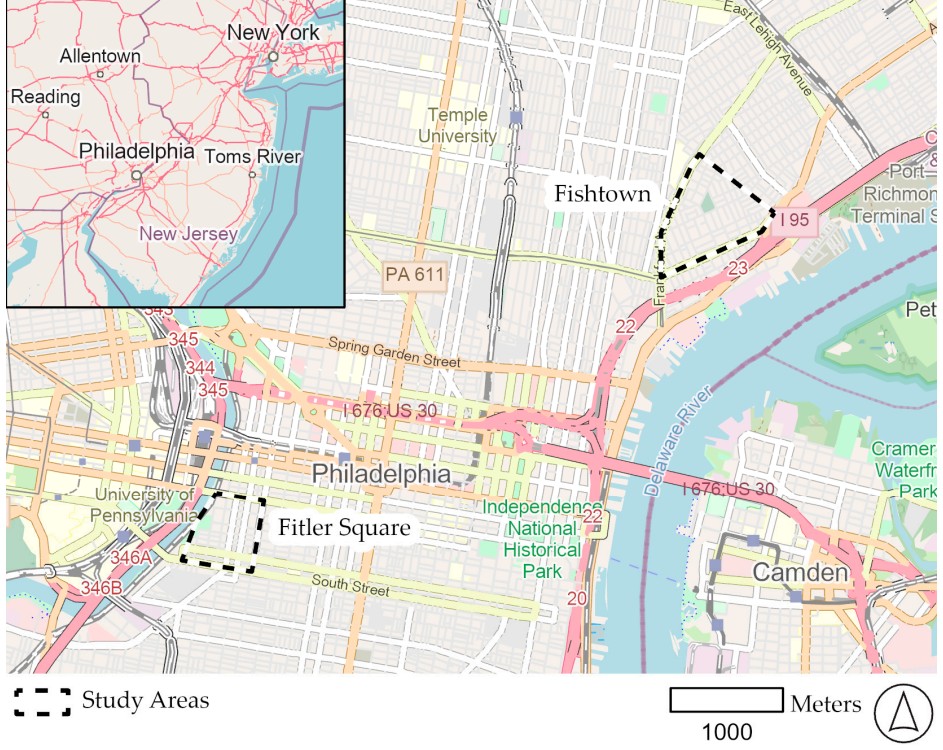

**Figure 3.** Fitler Square and Fishtown neighborhood study areas, Philadelphia, USA. Source: author created map using Esri's base map background.

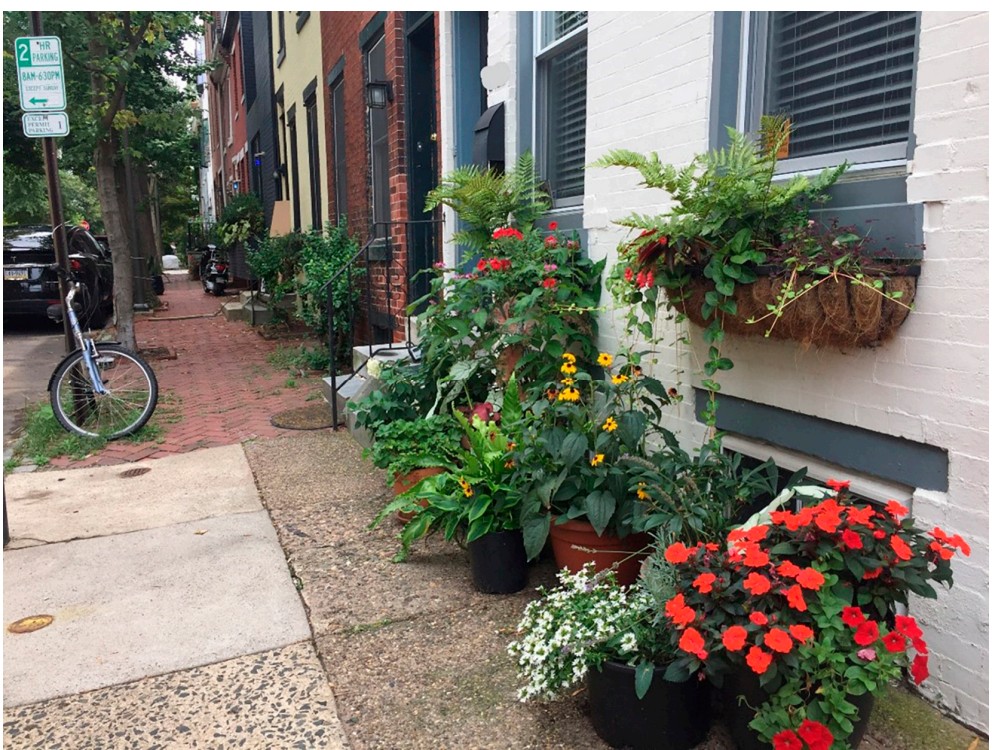

**Figure 4.** Streetside gardening in Fitler Square, Philadelphia (photograph by the authors).

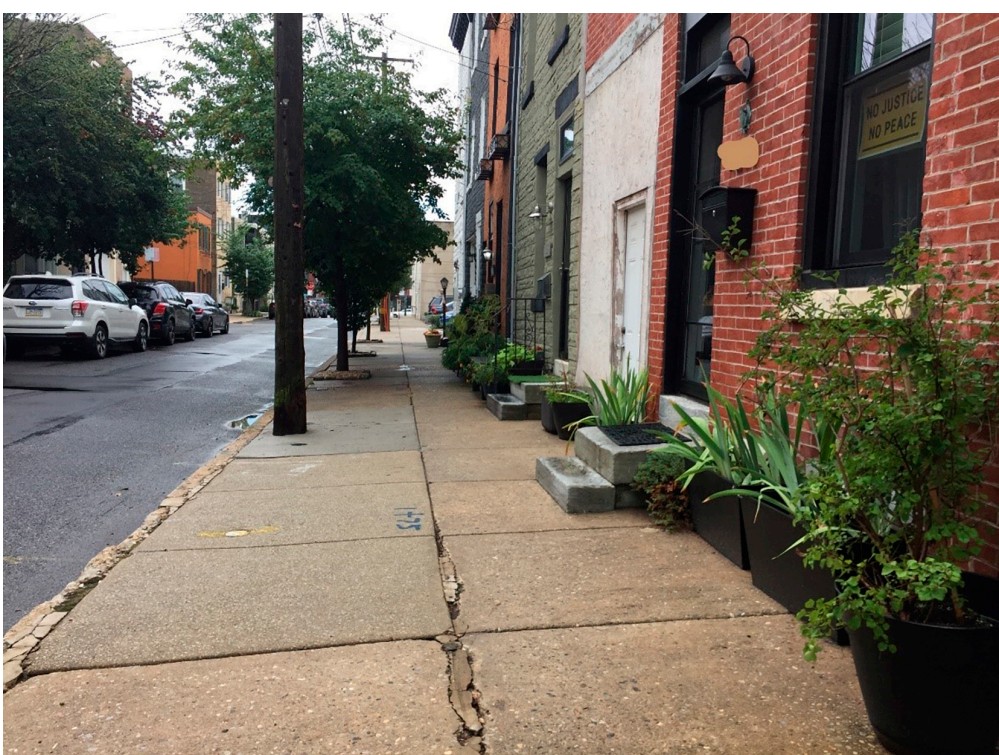

**Figure 5.** Streetside gardening in Fishtown, Philadelphia (photograph by the authors).

We compiled socio-economic and demographic data for Fishtown and Fitler Square from the EJScreen [23] and the Demographic Statistical Atlas [24,25] and for Delft's from City Population [26] and All Charts [27]. Additionally, all three study neighborhoods have similar population densities (Fitler: eastern sub-neighborhood 14,476/km$^2$ and western sub-neighborhood 9914/km$^2$; Fishtown: 10,656/km$^2$, and Central Core of Delft: Centrum

West 9820/km$^2$ and Centrum Southeast 12,000/km$^2$). All three neighborhoods are socio-economically and culturally mixed, middle-class communities and, thus, roughly comparable.

Both Fishtown and Fitler Square are currently majority-white neighborhoods (around 70–75% white) that have historically been working-class communities adjacent to industrial use sites. Fishtown still has a sizable working-class population but is seeing social and economic shifts toward a middle-class population. According to the Environmental Protection Agency (EPA) EJScreen Community Report for these two U.S. neighborhoods, owner-occupation of homes in these areas ranges between 42 and 58%. Most buildings in Fishtown and Fitler Square are row-house-style structures approximately 2.5 to 3 stories high. Fitler Square is approximately 225 hectares in area. Ninety-nine 100-m segments were surveyed both on foot and via Google Earth Pro. Fishtown is approximately 120 hectares in size. One hundred eighty-five 100-m segments were surveyed both on foot and online via Google Earth Pro.

Of Delft's overall 2022 population of just over 106,000, the Binnenstad ("city core") population was approximately 13,250. Foreign-born residents comprise 25.8% of the city's 2022 population [26]. The survey area for Delft was approximately 87 hectares in size, with one-hundred ninety-eight 100-m street segments reviewed via Google Earth Pro. Most buildings in this area are row-house-style structures approximately 2.5 to 3.5 stories high.

### 2.2. CGI Index Development

In constructing the CGI method, our initial data collection efforts were confined to Type 1 residential streets with either no other land uses or a very minor mixed commercial component. A full convivial greenstreet (CG) typology that included non-residential and celebratory greenstreet activities was given by Tamminga et al. in 2020 [22]. Based on the pilot CGI that followed, we were confident it could be easily adapted for the full range of CG types. Data collected included street name, orientation, type, plant sizes for left- and right-side or cardinal directions, plant containment method, segment length (30 m; explained below), segment aggregates, weighted aggregates, and notes on general observations (Figure 1). Information on basic morphology (i.e., height/width), and date of imagery were also recorded for later analyses and to be able to track changes through time.

To construct the CGI pilot method, plant units were differentiated by small, medium, and large size categories, as well as by the left/right side of the street for the three neighborhoods noted above. We considered small plant units to be uniquely identifiable as separate units, which were very small, solitary window-sill pots, mini wall-sconce pots, or in-ground plantings (including, notably, solitary voluntary plants). Small plant units had an approximate size of fewer than three liters (approximately the size of an adult human head) but larger than 1 US Cup or ~0.25 L (roughly the size of an orange). Medium plant units were solitary or groups of plants in pots or in-ground that were larger than 3 L (approximately the size of an adult head) and smaller than the average adult-sized non-cargo bike (0.8 m × 1.8 m × 1.2 m (30″ × 72″ × 48″)). Additionally, medium plants could also consist of plant masses where individual plants were no longer identifiable or a pot containing multiple species of plants. Large plant units were individual plants (pots or in-ground) or plant masses that exceeded the dimension units of medium plant units. Street trees installed by civic authorities were not included in the inventory; however, resident-initiated and resident-managed plantings within the tree pits were recorded. The plant collections' sensory and social impacts were not considered when factoring in plant size and form into the calculations for the CGI index instrument.

In comparing side-by-side photo-collages of sample street segments, we observed that the approximate biomass of three small plant units was similar in total spatial coverage to one medium-sized plant and that five small plants were about equivalent in coverage to a single large plant, as defined above (Figure 6). In calculating CGI for a given 30 m street segment, small plant units (spu) were 1* the counted number of spu, medium plant units

(mpu) were 3* the counted number of mpu, large plants (lpu) were 5* the counted number of lpu, and the street segment length was 30 m. This resulted in the following equation:

$$\text{Segment CGI} = \text{total } spu + 3*\text{\# of counted } mpu + 5*\text{\# of counted } lpu)/30 \text{ m} \tag{1}$$

Note that, in the formula, to normalize plant counts across plant sizes: 1 spu = 1 spu, 3 spu = 1 mpu, and 5 spu = 1 lpu. Using Figure 6 as an example results in

$$\text{CGI} = 2 + (3 \times 3) + (5 \times 2)/30$$

$$\text{CGI} = 2 + 9 + 10/30 \tag{2}$$

$$\text{CGI} = 0.7$$

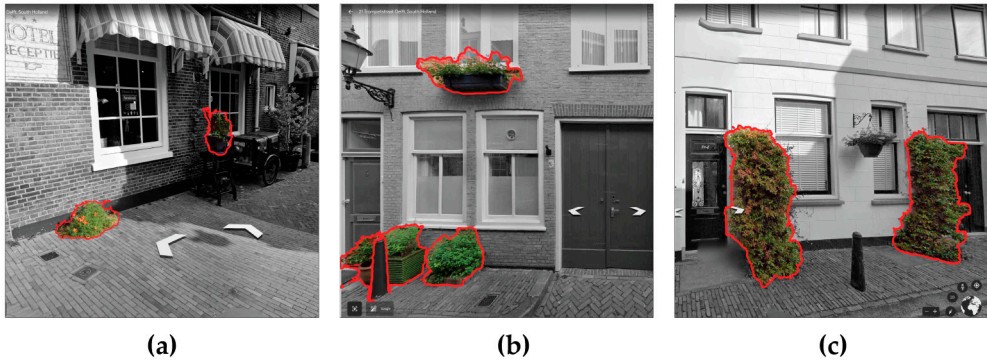

**(a)**            **(b)**            **(c)**

**Figure 6.** GSV screen captures highlighting the three plant unit size classifications and their relative sizes: (**a**) two spu, (**b**) three mpu, and (**c**) two lpu. Source: author adapted Google Street View image.

Thus, the CGI Index for an entire street, or even the entirety of Type 1 streets in a subject neighborhood, was

$$\text{Street or Neighborhood CGI} = (\text{total } spu + 3 \times \text{total } mpu \\ + 5 \times \text{total } lpu)/(\text{total Type 1 street length}/30 \text{ m}) \tag{3}$$

### 2.3. Street Rating

Streets within the study area were identified and mapped using Google Earth Pro and divided into 30-m segments. A 30-m street segment was chosen as the primary spatial unit for analysis. This segment size is approximate to the collection of greenstreet features that an investigator could visually absorb looking longitudinally down a streetscape setting in both online in Google Street View (GSV) and in-person along the street. This segment length also allowed for a sufficiently fine-grained analysis of streetside gardening within single street blocks while enabling patterns to emerge at the scale of most western European city neighborhoods. As shown in Figure 7, Type 1a and 1b streets in Delft's core ranged from three to 10 segments per street. Typical segments per street in Fitler Square and Fishtown neighborhoods were similar, except for several long Fishtown streets with over 30 segments. Each segment was assigned a street ID and unique segment code.

All streets addressed in this CGI pilot study were classified as Type 1a and 1b greenstreets, per Tamminga et al. [22].

A plant unit (*pu*) count was completed for every 30-m segment of the street. (Note that in Delft, lanes along both sides of a wide canal street were treated as their own separate streets since a pedestrian would likely experience each lane as a separate streetscape.) However, we used slightly different data-gathering techniques for each city as a way of exploring efficiencies in time and travel. In Delft, we began with core area visual reconnaissance on foot along all Type 1a and 1b streets (that is, streets with predominantly residential land use). Sample photography and field notes took place only on a few select

streets that appeared most densely vegetated. Thus, post-site work CGI Index construction for Delft relied heavily on recent Google Street View (GSV) imagery. For our Philadelphia study areas, all Type 1a and 1b streets were photo-inventoried in-person and on foot, along with accompanying field notes. Later, GSV imagery was used only to corroborate site photographs and double-check plant locations. Since Philadelphia had substantially more street parking than Delft—thus obscuring plants in GSV imagery—the in-person photo inventory was necessary to attain accuracy in plant counts.

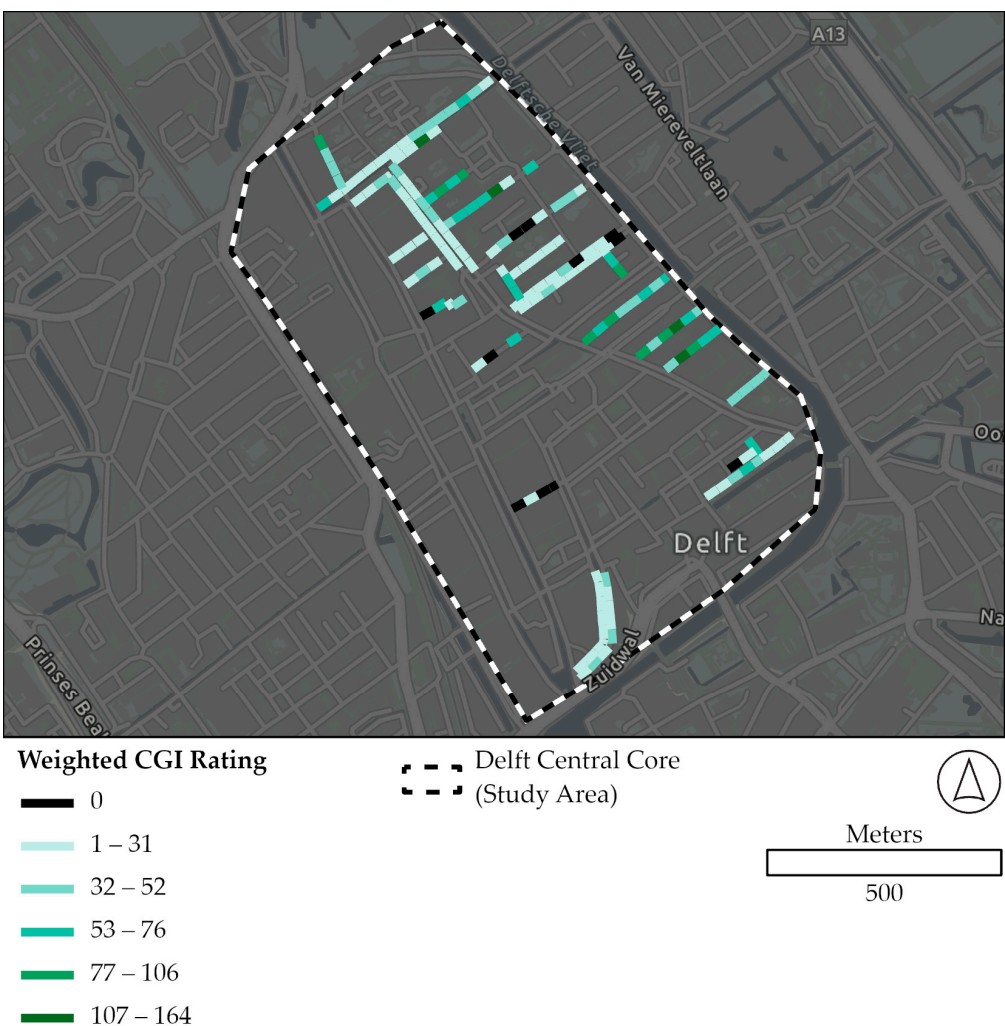

**Figure 7.** CGI Index map of CG activity in Delft Central Core by weighted 30 m segment aggregate. Source: author created map using Esri's base map background.

During neighborhood analyses, sample GSV screen captures were recorded along each street to document how the plant classification was being applied to different plant units (Figure 8). These screen captures were used to create a CGI Assessment Protocol (Appendix A) to train additional raters for validating the CGI Index and associated training protocols via an additional step of inter-rater reliability, described below.

The street segment data created in Google Earth Pro were exported as kmz files, which were then converted to a layer file in ArcMap. Finally, the survey data (i.e., plant unit class and size, street type, etc.) recorded in an Excel table were linked to the street segments in ArcMap based on the unique street segment code for mapping rating scores.

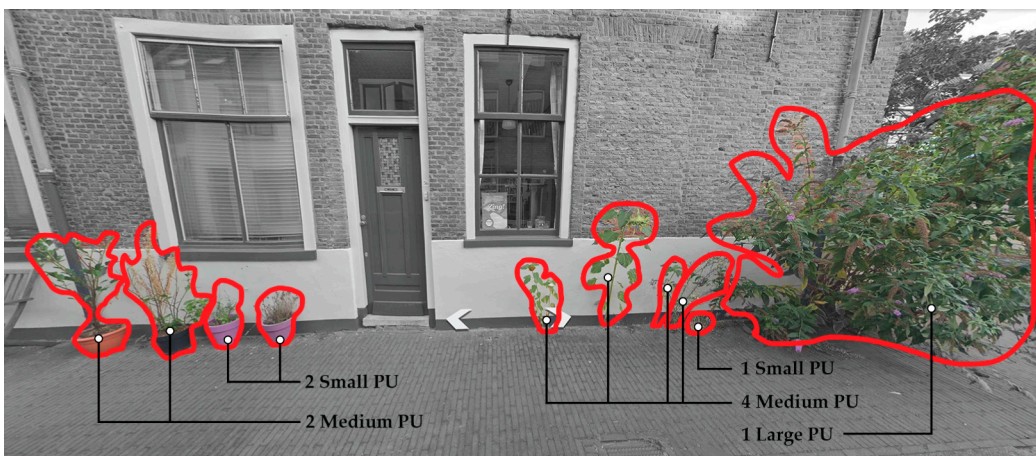

**Figure 8.** Example CGI Index screen capture showing a count for a portion of 56 Doelenstraat, Delft's left side street segment: three small, six medium, and one large plant units. Source: author adapted Google Street View image.

### 2.4. CGI Index Validation

A single rater on the research team initially rated all 516 street segments in both cities to limit interrater reliability concerns for street and city CGI score comparisons. However, we also wanted to ensure different raters would reproduce the initial CGI scores to allow for the expansion of city comparisons by multiple researchers over time and space. To test the reproducibility of the CGI assessment, we underwent a series of rating, training, and interrater reliability exercises [27] that led to the development and refinement of a CGI Assessment Protocol. In total, the three authors ranked 8 street segments two times. Between the first and second rankings, we developed a more refined CGI Assessment Protocol (see Appendix A). After the first round of ratings, we held a training meeting to refine the CGI Assessment Protocol further. We calculated interrater reliability (IRR) statistics after each rating session (see Section 3.2). The authors used the intraclass correlation coefficient (ICC) to evaluate rater agreement because the variables of interest were a continuous number of plant counts and CGI scores [28]. We calculated ICC for both plant size counts and CGI score on each street segment and reported the total CGI street score. We use Koo and Li's [28] IRR interpretation categories to report our IRR outcomes in the results section: kappa scores below 0.50—poor, between 0.50 and 0.75—moderate, between 0.75 and 0.90—good, and above 0.90—excellent. We used the r psych package [29] for all ICC calculations.

### 3. Results and Discussion

### 3.1. CGI Mapping

All greenstreets showed a wide range of plant growth forms and sizes. These included resident-grown vegetation on facades, along foundation walls where pavers were removed to create small foundation planters, where pots were nestled, and in tree pits along the street.

Maps of CGI-indexed streets in Delft's core and the two Philadelphia neighborhoods are shown in Figures 9 and 10. Streets with a score of zero were included because, although they were essentially plantless at the time of GSV photography (Delft) or on-site inventory (Philadelphia), they fit the criteria of Type 1a or 1b street type and served as a convenient baseline. Type 2 streets (largely commercial land uses) were excluded. Table 1 shows the totality of the study areas' CGI-indexed streets.

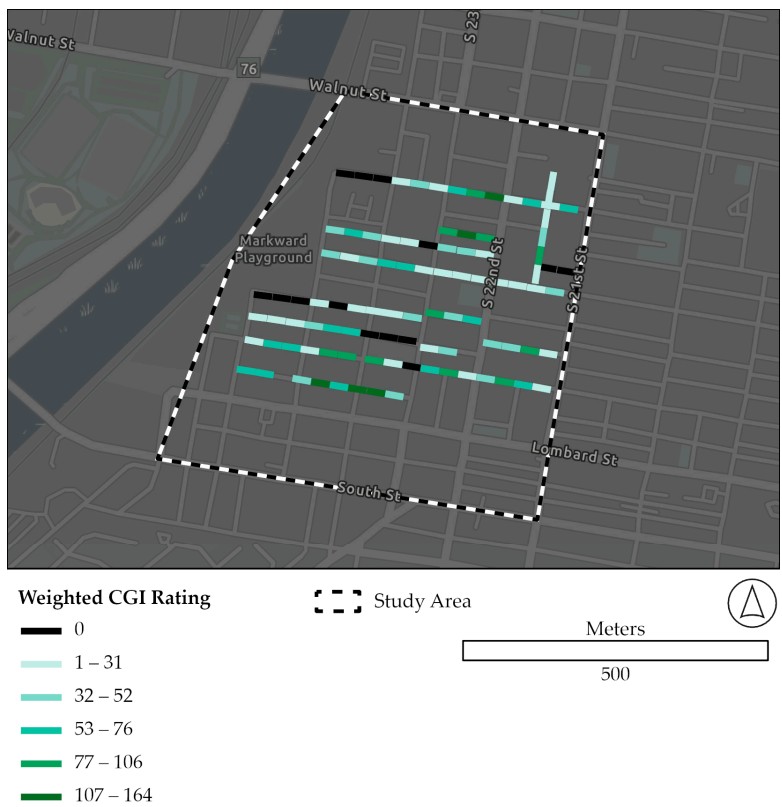

**Figure 9.** CGI Index map of CG activity in Fitler Square by weighted 30 m segment aggregate. Source: author created map using Esri's base map background.

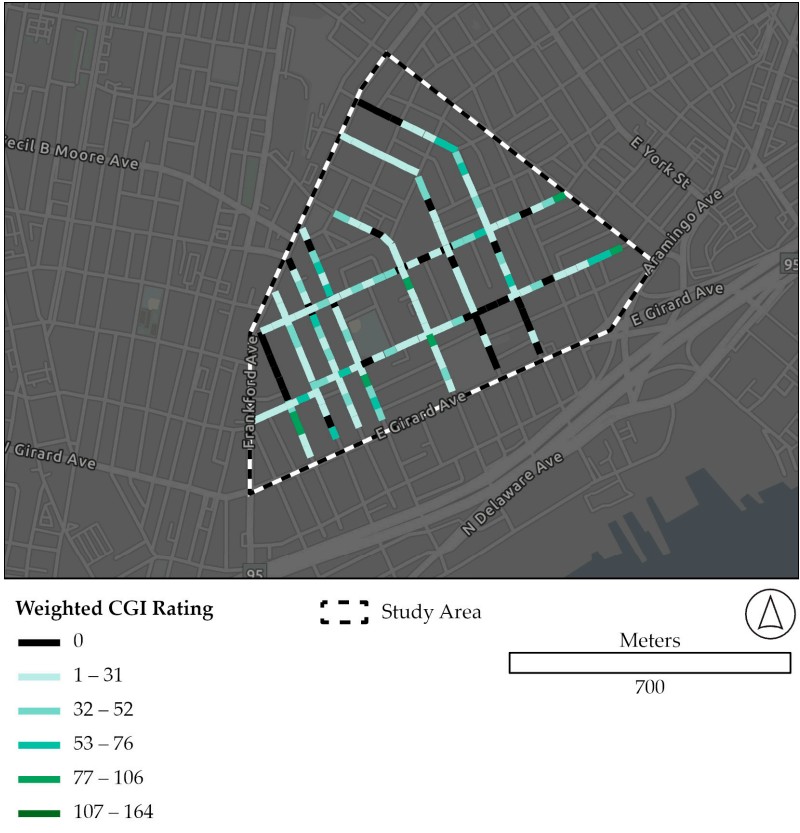

**Figure 10.** CGI Index map of CG activity in Fishtown by weighted 30 m segment aggregate. Source: author created map using Esri's base map background.

**Table 1.** Ranking of CGI-indexed streets for Delft and Philadelphia study areas. * Source: own work.

| Street Name | Average Weighted CGI (WPU/30 m) |
|---|---|
| Rittenhouse | 101 |
| Waverly | 89 |
| Van der Mastenstraat | 79.25 |
| Harmenkokslaan | 78 |
| Doelenstraat | 63 |
| Donkerstraat | 62.57 |
| Houthaak | 62 |
| Trompetstraat | 61 |
| Pine | 55.69 |
| Vlamingstraat | 54.62 |
| Annastraat | 51.75 |
| Gasthuislaan | 51.11 |
| Pluympot | 48.33 |
| Oranjestraat | 47.75 |
| Verwerskijk | 46.55 |
| Locust | 40.31 |
| Achteron | 37.62 |
| VanPelt | 34.17 |
| Minderbroestra | 33.75 |
| Kantoorgracht | 33.5 |
| Columbia | 33 |
| Zuiderstraat | 32.25 |
| Delancy | 31.88 |
| Geerweg | 31 |
| Spruce | 30.54 |
| Rietveld | 29.88 |
| De Vlouw | 28 |
| Panama | 26.5 |
| Fortuinstraat | 26 |
| Molslaan | 26 |
| Palmer | 25.89 |
| Belgrade | 25.44 |
| Manning | 24.36 |
| Thompson | 23.67 |
| Crease | 23.36 |
| Molenstraat | 22.52 |
| Drie Akersstraat | 22.33 |
| Visstraat | 22 |
| Oxford | 21.27 |
| Marlborough | 21.08 |
| Berks | 20.56 |
| Montgomery | 13.62 |
| Schutterstraat | 13.6 |
| Vaandelstraat | 11.6 |
| Molstraat | 4 |
| Vijverstraat | 3 |
| Spieringstraat | 1.12 |
| Huyterstraat | 0 |
| Klooster | 0 |
| Breestraat | 0 |
| Smitsteeg | 0 |

* blue = Delft, purple = Fishtown, red = Fitler Square.

In concert with spatial mapping, the utility of CGI indexing becomes readily apparent. Our early assumption had been that Leiden's core would not only be more consistently graced with CG (this assumption was confirmed) but also have the highest individual street scores. In fact, several of the most robust greenstreets were found in the Fitler Square

study area. Other interesting spatial features begin to be revealed in the CG intensity mapping. For instance, all three study areas showed evidence of CG clustering at the sub-neighborhood scale. This suggests the possibility of a spatial contagion effect, as discussed for other forms of urban gardening in Hunter and Brown [30]. It also begins to help target focused sociological and green infrastructure research on the robust CG nodes within a broader study area.

### 3.2. ICC IRR

The intra-class correlation coefficient (type ICC2) was computed to assess the agreement between three raters in counting plants for eight street segments (note: each of the eight street segments was divided into east and west sides of the street for a total of 16 comparison ratings) (Table 2). Additionally, the CGI rater reliability was also calculated for a complete street segment (Table 3).

**Table 2.** ICC IRR results for each east and west street segment per plant size counts per rating round. Source: own work.

| Rating Round | Segment | Rating | Kappa | p |
|---|---|---|---|---|
| Round 1 | West_Small | Poor | 0.0000000 | 0.4706248 |
| Round 2 | West_Small | Moderate | 0.7333899 | 0.0002487354 |
| Round 1 | West_Medium | Moderate | 0.7448085 | $9.296207 \times 10^{-6}$ |
| Round 2 | West_Medium | Excellent | 0.9799844 | $2.224643 \times 10^{-13}$ |
| Round 1 | West_Large | Good | 0.7906977 | $5.034793 \times 10^{-5}$ |
| Round 2 | West_Large | Good | 0.8383377 | $8.923815 \times 10^{-6}$ |
| Round 1 | East_Small | Poor | 0.3741546 | 0.01841982 |
| Round 2 | East_Small | Moderate | 0.6043832 | 0.0003941142 |
| Round 1 | East_Medium | Poor | 0.3389165 | 0.02897330 |
| Round 2 | East_Medium | Moderate | 0.7002510 | 0.0002027596 |
| Round 1 | East_Large | Moderate | 0.6543219 | $2.719121 \times 10^{-5}$ |
| Round 2 | East_Large | Good | 0.7606796 | $1.223398 \times 10^{-4}$ |

**Table 3.** ICC IRR rating scores for each street segment CGI score. Source: own work.

| Rating Round | Rating | Kappa | p |
|---|---|---|---|
| Round 1 | Moderate | 0.5433127 | 0.0005696958 |
| Round 2 | Excellent | 0.9460622 | $4.855641 \times 10^{-9}$ |

Finally, The CGI score for Trompetstraat for each rater was as follows: round one, 431.4196, 548.4196, and 372.1071; and round two, 507.0714, 515.7500, 527.7500. The spread of the CGI scores decreased from 176.3125 to 20.6786 from round one to round two.

### 3.3. Discussion

Constructing the CGI index allowed spatial characteristics of greenstreets in Delft and Philadelphia to be mapped and visually assessed to reveal emerging patterns of CG intensity and distribution. They also invite ready comparisons of CG activity between CG clusters, sub-neighborhoods, neighborhoods, and cities.

Parking and loading regulatory framework, streetscape management, construction operations, and level of on-street enforcement on the day that GSV imagery occurred can all impact the suitability of GSV imagery for creating a CGI index. Lower amounts of street parking in Delft allowed for relatively accurate plant counts using GSV imagery. However, large amounts of street parking and ongoing construction in some parts of the Philadelphia neighborhoods made remote GSV imagery a less accurate data source—thus necessitating on-site photo inventory by the researchers. Further surveys will need to balance the resource and time-saving advantages of using GSV imagery over potential losses in accuracy.

Because sidewalk-based installations were sometimes obscured by parked cars or delivery vehicles in Google Earth imagery, the on-site photography gathered during earlier reconnaissance was occasionally checked to confirm uncertainties. Whether a survey relies primarily on GSV imagery or in-person data collection, it can be helpful to have the other data collection strategy as a secondary source.

The CGI Index approach described above tries to accommodate for the inherent subjectivity of sizing plants and aggregating plant massings along the street. While assigning weighting factors was based on our best judgment in two temperate climate cities, future researchers may make adjustments based on their urban contexts and the level of accuracy needed to address the research question. For example, research on arid cities or tropical cities may develop plant unit weightings that better suit those settings.

## 4. Conclusions

As noted earlier, there is a dearth of research on greenstreets that are characterized primarily by informal streetside gardening, and to our knowledge, there has been no work on CG spatial characteristics or CG activity intensity levels in particular. We offered a definition of this urban phenomenon and established it within larger sets of urbanistic theories. Next, we proposed a way to quantitatively document greenstreet spatial extents and levels of streetside gardening intensity as a necessary prelude to further exploration of potential CG ecosystem services and social variables. We have shown above that the integration of widely available ESRI-based ArcMap and Google Earth tools and associated data sets is poised to help advance the expanding greenstreets research agenda. With the CGI Index approach, we anticipate that the collection of greenstreet plant unit data via ArcMap and GSV will be much more efficient than collecting the same amount of data on site. Savings in money spent on fieldwork and travel, as well as carbon footprint reduction, could be expected.

The development and application of the CGI Index approach revealed several limitations. First, StreetView imagery in Google Earth does not always reliably provide full visual access to streetside activities; indeed, we needed to resort to in situ photographs in Philadelphia to augment StreetView images that were obstructed by parked delivery vehicles. Relatedly, CG changes between GSV photo-panoramic image dates and the present time can vary; future researchers may wish to investigate the impact of differing imagery and active research dates. Finally, it soon became evident that a post-CG inventory phase of interrater reliability was essential in reducing subjectivity in assessing CG plant sizes and assigning scores. While we conducted an initial high inter-rater reliability test for CGI Index measures of a single sample street (Trompetstraat) in Delft, this step should be adopted as a norm in future CGI work. For this single research venue, we found that data replicability became achievable only after consensus-building training amongst the research team.

More broadly, future work on CG metrics could include factors linking the characteristics of streetside plants with spatial and temporal metrics at the street (e.g., cross-sectional street width-to-building height) and neighborhood scales (e.g., floor area ratio; other density measures). CGI can also play a role in future work across a spectrum of urbanism, ecology, engineering, and public policy inquiries in addressing questions on the interplay between greenstreets and urban spatial morphology, such as sidewalk/street widths and façade articulations. Finally, we see good potential in using CGI Indices as a primary spatial data set in investigating links between levels of convivial greenstreet activity and urban microclimatic and physical and mental health. Ultimately, methodical approaches to convivial greenstreets, such as those presented herein, should help establish more supportive contexts for streetside gardeners to practice their craft and collectively enhance the urban environment for all.

**Author Contributions:** Conceptualization, K.T. and E.K.; methodology, E.K., K.T. and T.F.; software, T.F.; validation, E.K., K.T. and T.F.; formal analysis, E.K. and T.F.; investigation, E.K., K.T. and T.F.; data curation, E.K.; writing—original draft preparation, E.K. and K.T.; writing—review and editing, E.K., K.T. and T.F.; supervision, K.T.; and project administration, E.K. All authors have read and agreed to the published version of the manuscript.

**Funding:** This research received no external funding.

**Institutional Review Board Statement:** Not applicable.

**Informed Consent Statement:** Not applicable.

**Data Availability Statement:** Data are available upon request.

**Conflicts of Interest:** The authors declare no conflict of interest.

## Appendix A

CGI Assessment Protocol

Opening Street Transects—Google Earth Instructions

(1)      Go to https://earth.google.com/web/ (accessed on 8 October 2023) and log in using your Google Account.
(2)      Open the menu (three bars in the top left of the screen (Figure A1) and click on projects.

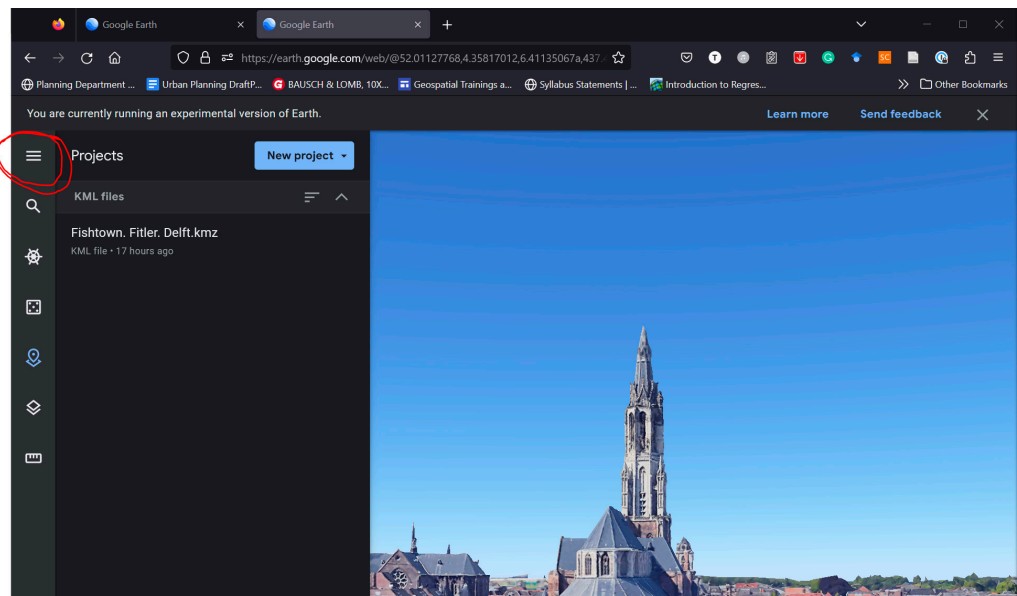

**Figure A1.** Menu location.

(3)      In the projects menu, click on Open > Import KML file from the computer.
(4)      Browse to the location of your Fishtown. Fitler. Delft.kmz and click open.
(5)      Open the nested folders (click on the arrows next to the folder in your project; see Figure A2) and then double-click on a segment to zoom into that segment.
(6)      To get to the Street View, drag the person in the lower right onto the street to be viewed.

Convivial Green Street Plant Unit Counting Instructions

(1)      Note that streets are marked with 100-m line segments in Google Earth Pro. Segments are named, and each street segment's plant per small, medium, and large category. Log the counts in a spreadsheet.

(2)  Drop into Street View on one end of the street in Google Earth Pro. "Walk" down the street for the length of one segment, counting the plant units on one side of the street. Plant counts should be recorded for small, medium, or large.

(3)  Once one side of a street segment is finished, return to the beginning of the street segment and count the plant units on the opposite side of the street.

(4)  Continue steps 3 and 4 for each 100-m segment down the length of the street.

(5)  Note:

(a)  We are not counting official/formal city street vegetation, such as street trees or city-maintained vegetation. We acknowledge this can be messy when determining planting providence. However, Figures A3 and A4 below illustrate the formality of plants not to count.

(b)  Additionally, the plant base must be located within the street side of the building facades. We will not count vegetation rooted in parklets or yards, even if it hangs over the wall into the streetscape. Figure A5 is provided below as an example.

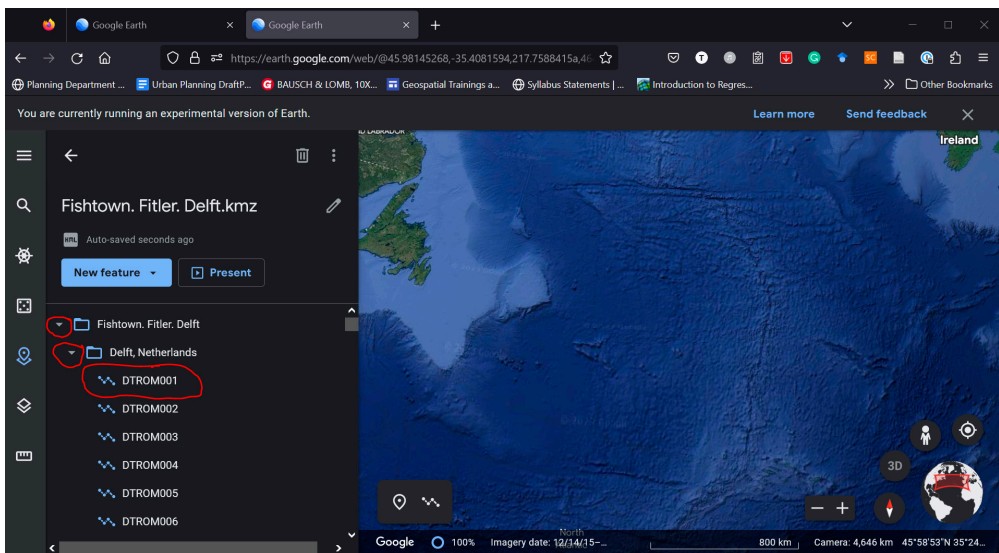

**Figure A2.** Nested folders and street segment.

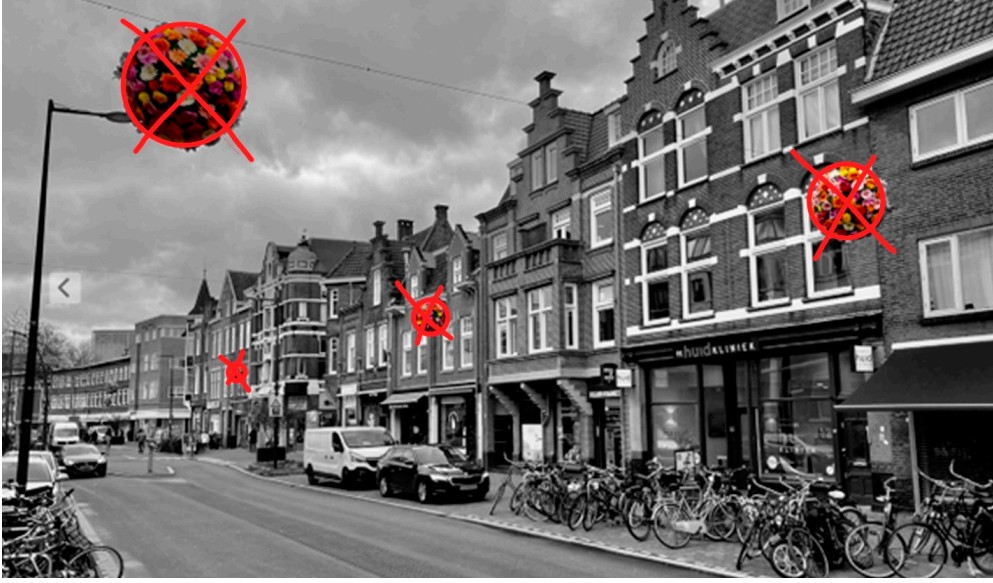

**Figure A3.** Do not count example 1.

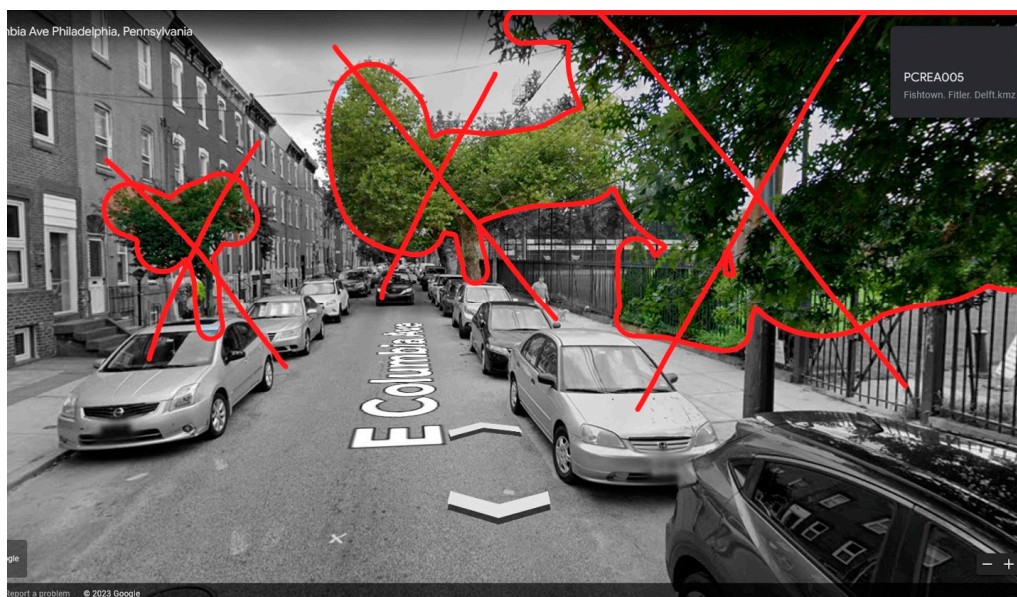

**Figure A4.** Do not count example 2. These plants do not count because they are street trees and vegetation beyond the streetscape façade, even though they grow through the fence.

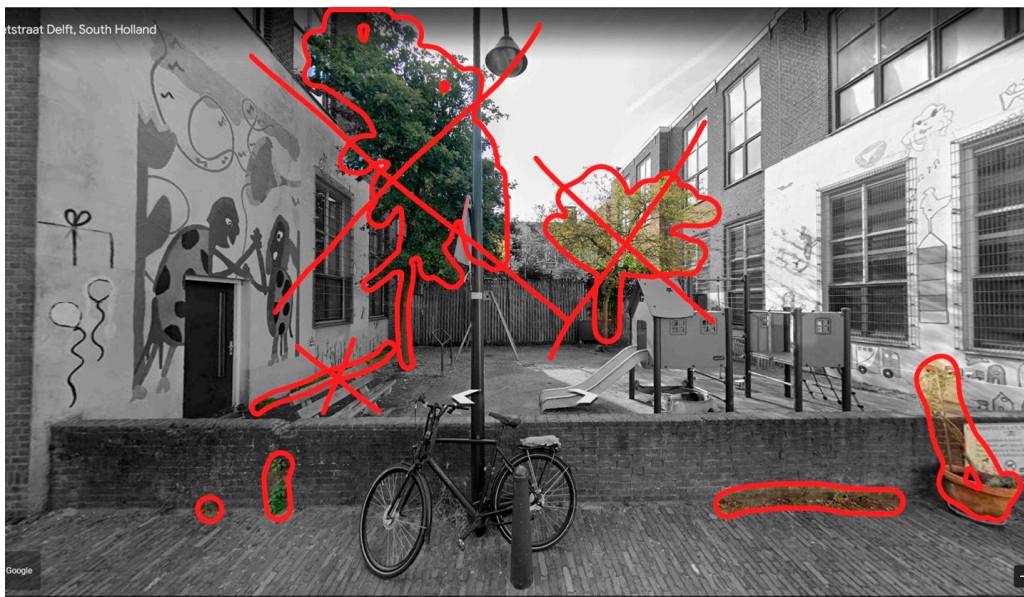

**Figure A5.** Do not count example 3. Plants with the red outline count (4), and plants with the red X do not count (3) because they are within a pocket park beyond the streetscape facade.

Plant Unit Definitions and Examples
*Small*

Small plant units are uniquely identifiable as separate units, very small, solitary window-sill pots, mini wall-sconce pots, or in-ground plantings (including, notably, solitary, voluntary plants). Small plant units shall have biomass or an approximate size of fewer than three liters, approximately the size of an adult human head, but larger than 1 US Cup or ~0.25 L or roughly the size of an orange. Figures A6–A8 show examples of small plant units.

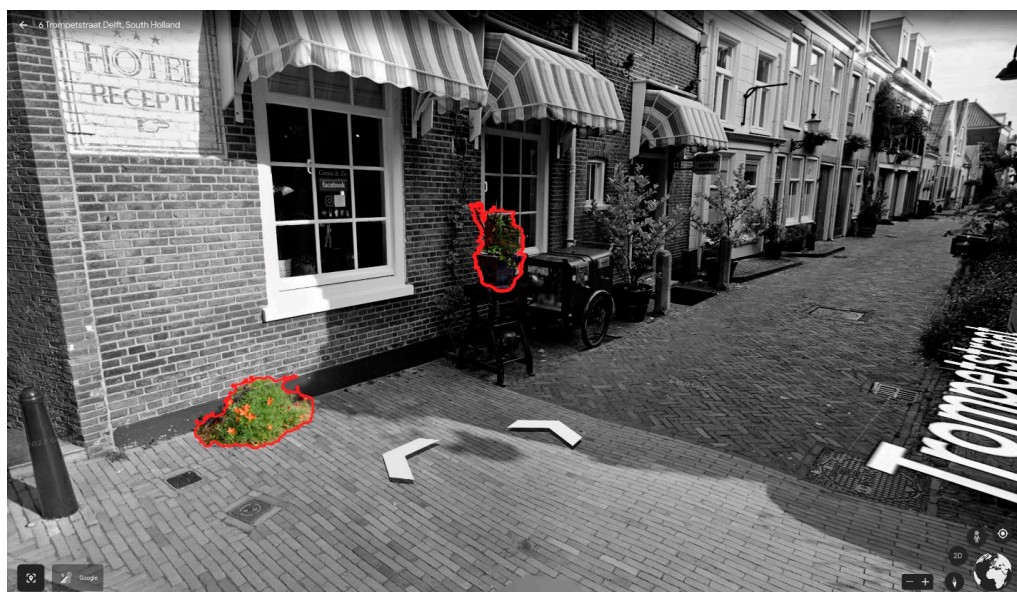

**Figure A6.** Small Example 1: two small plant units.

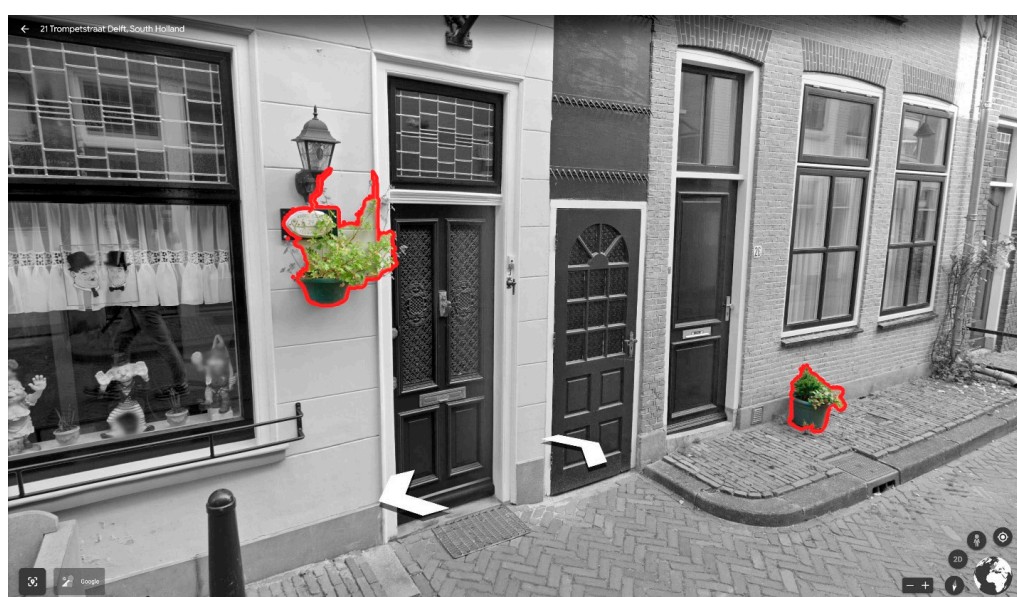

**Figure A7.** Small Example 2: two small plant units.

*Medium*

Medium plant units are solitary or groups of plants in pots or in-ground that are larger than 3 L (approximately the size of an adult head) and smaller than the average adult-sized (0.8 m × 1.8 m × 1.2 m (30″ × 72″ × 48″)), individual, non-cargo bike, but can also consist of plant masses, where individual plants are no longer identifiable or a pot containing multiple species of plants. Note that the plant should not exceed the height of the bike dimension listed above; if it does, it shall be classified as a large plant. Finally, this can include small shrubs, trees, and vines if they do not exceed the bicycle dimensions listed above. Figures A9 and A10 are examples of medium plant units.

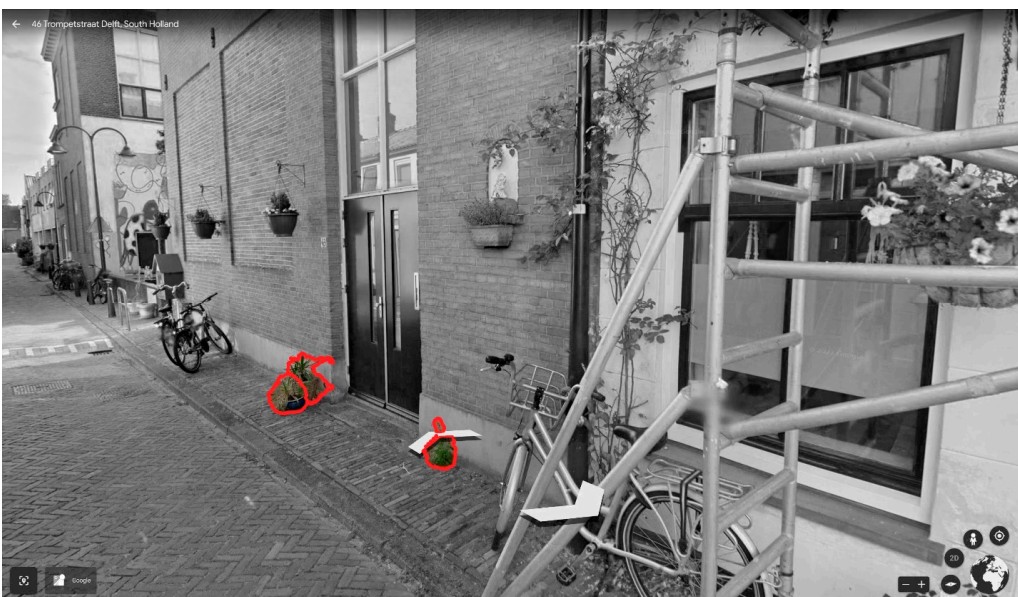

**Figure A8.** Small Example 3: three individual small plant units.

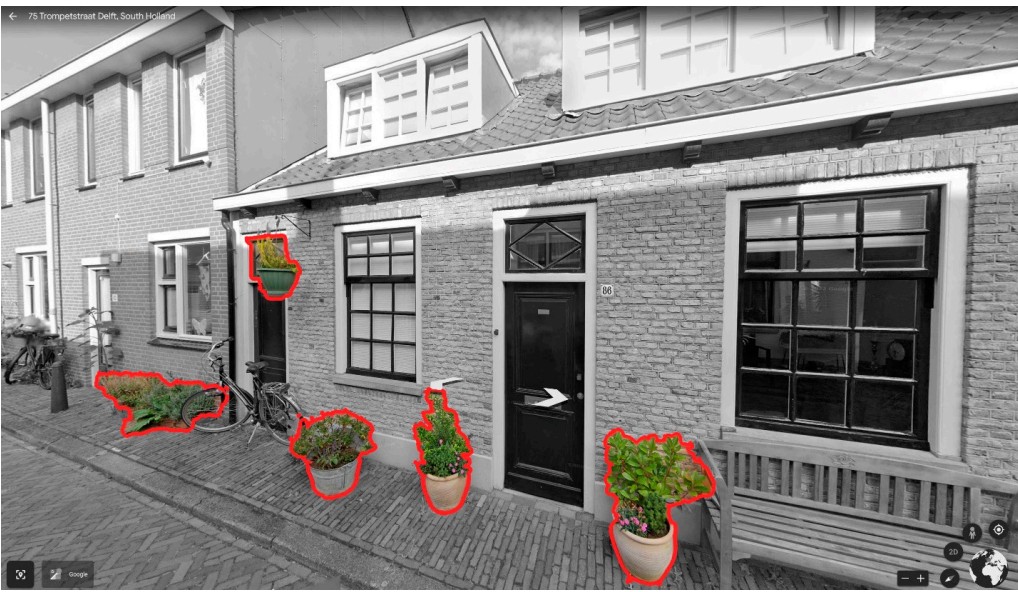

**Figure A9.** Medium Example 1: five medium plants. Note the grouping of in-ground planted plants by the bike tire is counted as a single, medium plant unit, while the sunflower in the photo's background is a large plant because it is taller than the average adult, non-cargo bike.

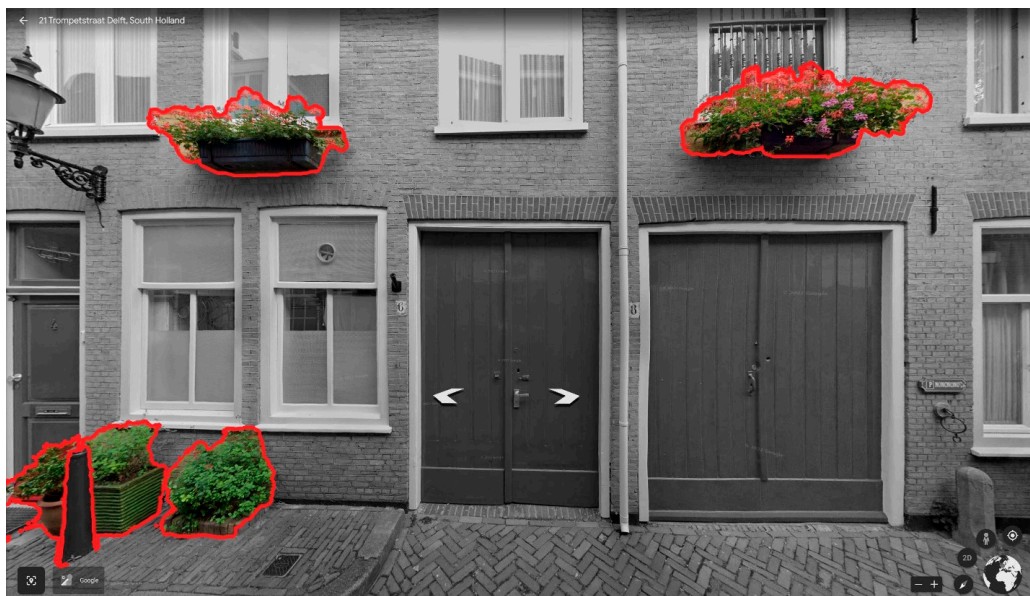

**Figure A10.** Medium Example 2: five medium plant units.

*Large*

Large plant units are individual plants (pots or in-ground) or plant masses that exceed the dimension units of medium plant units (see above). Figures A11–A13 are examples of large plant units.

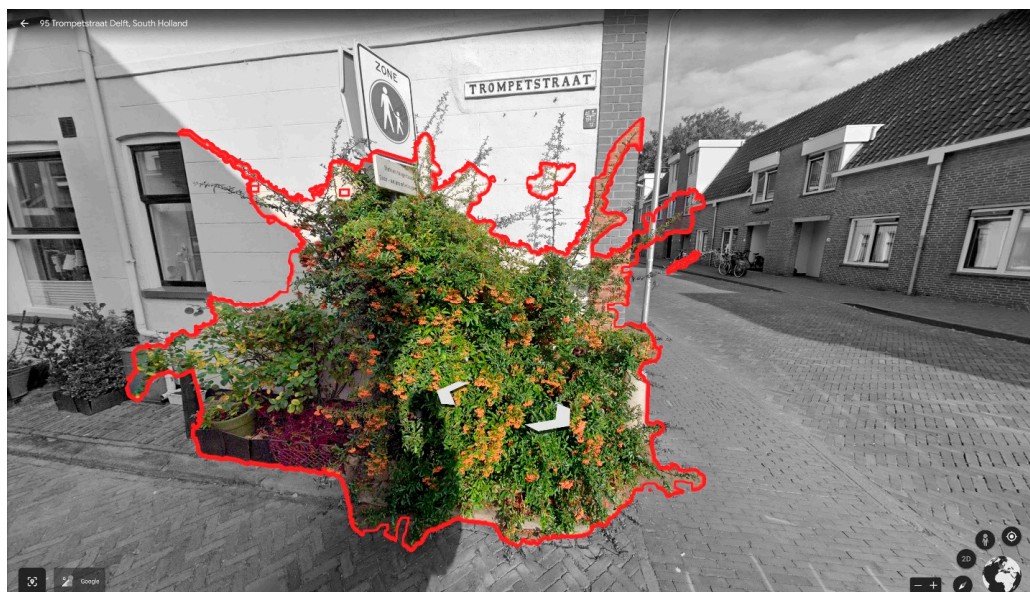

**Figure A11.** Large Example 1: 1 large plant mass counts as 1 large plant unit.

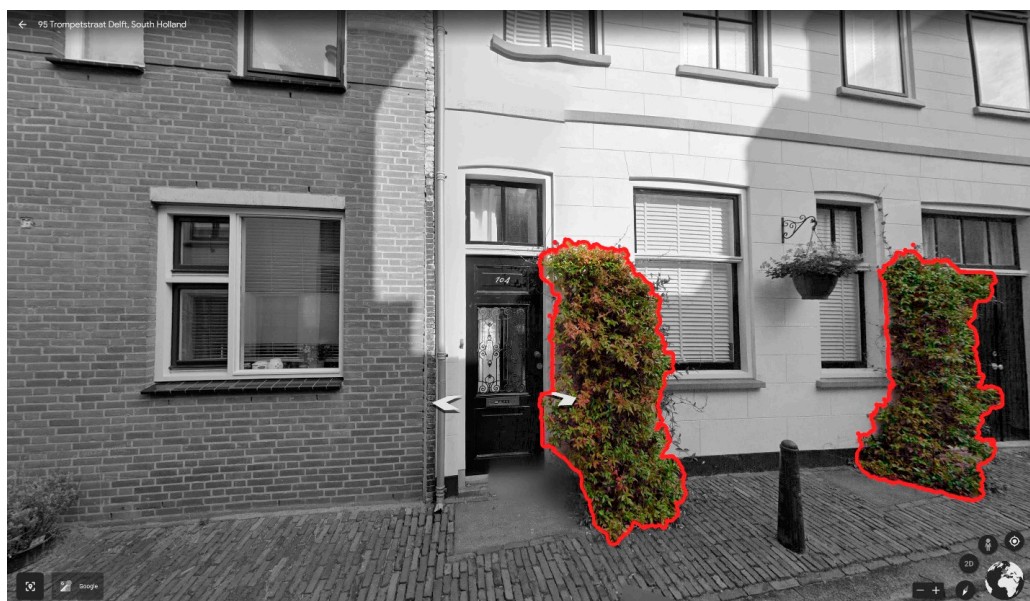

**Figure A12.** Large Example 2: 2 large plant units.

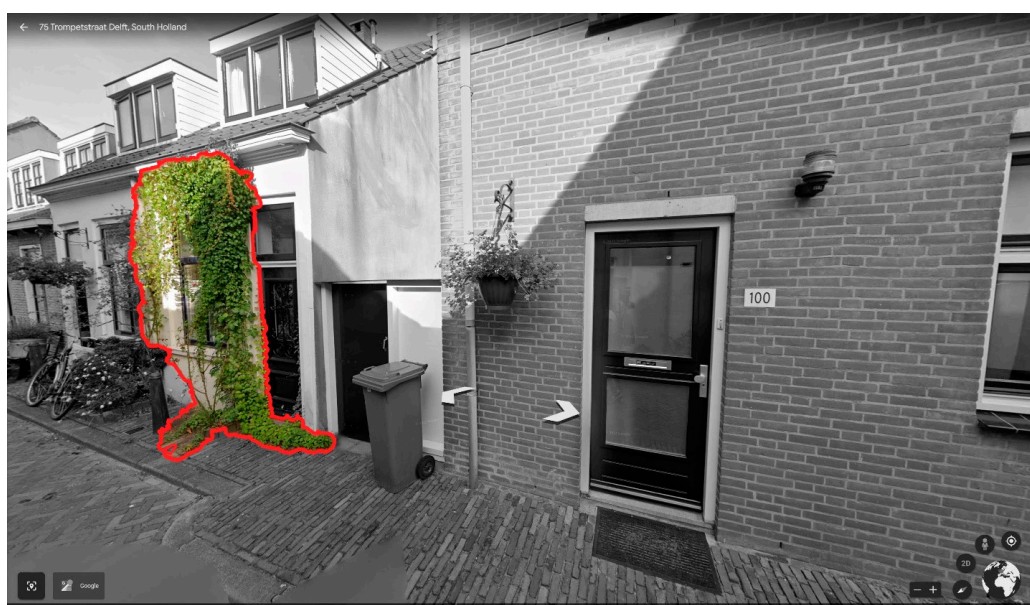

**Figure A13.** Large Example 3: 1 large plant unit.

Additional Complex Plant Count Examples

　　Figure A14 documents how to count plants in complex situations.

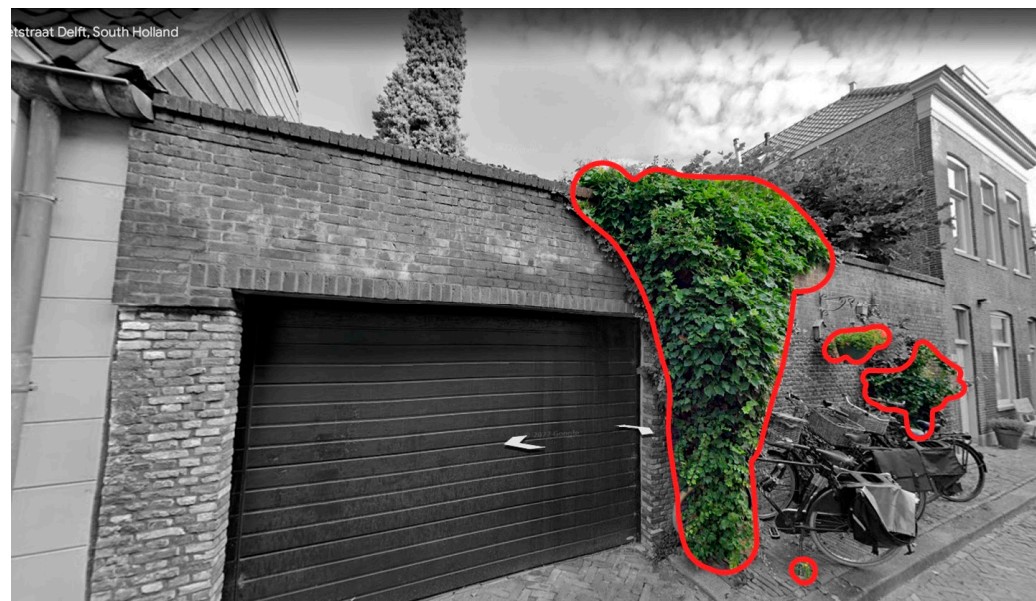

**Figure A14.** Complex plant count 001: The count for this image is 1 small, 1 medium, and 2 large plant units. Note that the two trees behind the wall are not counted, as they are rooted behind the streetscape façade.

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
