# Peer review of "Spatial Indices for Convivial Greenstreets"

_sustainability, doi:10.3390/su152416781_

Round 1

Reviewer 1 Report

Comments and Suggestions for Authors

Comments on Manuscript ID: sustainability-2699533-peer-review-v1

Title: Investigating the Spatial Characteristics of Convivial Green- 2 streets: The CGI Index

Comments

The study will have a substantial contribution to the field if the authors revise the manuscript. The overall manuscript structure is not good, and the write-up needs further work. Thus, for further improvement, I would like to forward the following comments in each sub-section of the manuscript:

Dear authors, I cannot understand the main intention of the study. Is the study designed to prove the suitability of ArcMap and Google Earth tools to assess green streets? Or assessing the green status of the three cities? With what criteria did you select those cities?

Specific comments

  1. Abstract
  1. The abstract seems good, but the authors missed including the findings of the study.
  2. Why too many keywords? Consider the maximum number of keywords in the author manual. 13 keywords are too many.
  1. Introduction

After reading the introduction section, I still cannot find out the knowledge gap and main innovation of this study. Also, the study objectives are not clearly stated. This section is not well organized to disseminate the required information.

The authors should consider the following issues:

  1. The novelty of the study is not articulated well.
  2. The study objectives are not clearly stated.
  3. There are too many direct quotations, which may reduce the reader's interest and seem like historical studies.
  1. The authors can avoid the short bio statement of the researchers they cited.

2.1 Study Area,

  1. The readers expect you to state the study areas, but there is no need to explain the data sources, like socioeconomic and demographic data.
  2. You can merge this with the materials and methods section.

2.2. Materials and methods

    1. Results
  • The way you state the results is good, but the result section is not adequate.
  • The legends in figures 6 and 7 are not readable.
    1. Discussion

This is not adequately discussed unless you can merge it with the results section under the heading Results and Discussion. Also, why not compare your work with other similar studies elsewhere? If this study is the first of its kind, inform the readers of that information.

  1. Conclusions

The conclusion is based on the results.

For minor comments, please see the athached pdf file

Comments on the Quality of English Language

Some long sentences are seen in the manuscript and need to be rewrite.

Author Response

Dear Reviewer 1,

Thank you very much for your insightful review. Please see the attached letter which details our responses to your comments.

Sincerely

Reviewer 2 Report

Comments and Suggestions for Authors

The article is interesting and has potential, but requires major corrections. The literature review is good, coherent and does not require additions. Thank you to the authors and congratulations. However, there are some details in this article that may require further review and careful checking. Here are some examples for reference.

1.       Could you please describe what is the message of this paper? What was the aim of the research? These are the main questions to clear for the readers.

2.       Authors must refer to the Sustainability Author Guidelines https://www.mdpi.com/journal/sustainability/instructions and write your manuscript according to the Microsoft Word template or LaTeX template. The reviewed article is not written in accordance with the guidelines.

3.       Please read the guidelines and adapt your article to these guidelines.

4.       Authors must refer to the Sustainability Author Guidelines https://www.mdpi.com/journal/sustainability/instructions and write references properly.

5.       Please present the Introduction at the end: What is the research gap?

6.       Please present the Introduction at the end: Finally, briefly mention the main aim of the work and highlight the principal conclusions.

7.       Please provide a diagram of the research methodology used

8.       In your final conclusions, please specify the limitations of the proposed study? What exactly do they mean for other researchers?

9.       Who is the decision maker in the study? Who will gain the maximum benefit from the article?

10.    Why did the authors leave a comment=question EK1 (lines 181-184).

11.    Please provide a diagram of the research method at the end of the Introduction or in section 3.

12.    Below line 213, please provide a block diagram for determining the CGI index.

13.    Please include these photo-collages of sample street segments in the article. It will be better to understand the method described.

14.    Please in article present formula (1) in the form of a graphic diagram to make it more understandable to the reader.

15.    In line 242, [] reference to Google Earth is missing.

16.    Please provide a graphic interpretation of the accepted assessment on line 258.

17.    Please also attach a graphical text visualization diagram to lines 276-279

18.    Authors should make a more clear connection between figures illustrating their method and the text

19.    Please describe the weight assignment more clearly

20.    The conclusions from the research are very good and promising, but the authors should take more care to clearly describe their method. Based on the poorly and unclearly presented content in the article, it will be difficult for any researcher to repeat the research, because the article lacks a step-by-step description of the proposed method and a clear presentation in the simplest block diagrams or visualizations.

Author Response

Dear Reviewer 2,

Thank you very much for your insightful review. Please see the attached letter which details our responses to your comments.

Sincerely

Reviewer 3 Report

Comments and Suggestions for Authors

In this paper, the authors propose to investigate the spatial characteristics Convivial Green streets by determining the CGI ( Convivial Greenstreet Intensity) index. They propose a three-stage methodology: creating the CGI Index, rating street segments and assessing the interrater reliability of the CGI Index. They presented the obtained results through maps.

While reading the text, I noticed some flaws that I think should be corrected before publication: Lines 174 and 177 the word hecatares should be replaced with hectares.

Figure 1 is completely uninformative. Almost complete whiteness allows more content to be displayed for a better view of the area. Given that the work uses the SI system of units, I suggest that the scale be displayed in kilometers.

Figure 2 shows the Delft area better, but the legend is missing. Also, the display is without a proper cartographic projection, so the shape of the city area is deformed. I suggest a scale in kilometers.

Figure 5. The map needs to be repaired, the legend and the scale are too small, the display is deformed. I suggest to apply UTM projection so that the display is not deformed.

Figure 6: legend hard to read, scale almost invisible. Display in UTM projection.

Figure 7. Enlarge legend and scale, display in UTM projection.

In general, for all maps, increase the legend and scale to make them readable. For the scale, replace miles with kilometers or add the scale with kilometers. Display the map in an adequate projection, because now the display is deformed, I suggest displaying it in UTM projection.

Author Response

Dear Reviewer,

Thank you very much for your insightful review. Please see the attached letter which details our responses to your comments.

Sincerely

Round 2

Reviewer 1 Report

Comments and Suggestions for Authors

On the title, replace the abbreviation CGI with the full term.

Author Response

Dear Reviewer 1,

Please find attached a cover letter that addresses the second round of minor revisions. You will find the revisions in blue text.

We very much appreciate your role in helping to improve our paper.

Reviewer 2 Report

Comments and Suggestions for Authors

Comments:

I would like to thank the authors for improving the manuscript and responding positively to the comments. I am convinced that the authors themselves see the difference between versions 1 and 2 of the article. Thank you especially for the methodology diagram. This will allow readers to understand the authors' message in the article more quickly and easily.

Unfortunately, there are still some things to improve:

1.       E.g. line 50. Not [2] (p.11), but [2] (p. 11). Space is missing. Check the Instructions for Authors.

2.       Below Figures 1 and 6, and perhaps also others and Tables 1, 2 and 3, please add at the end: Source: own work.

3.       Are the equation and Figure 6 good? This means that 3 small = 1 medium and 5 small = 1 large. So on a 30 m section: 1 small + 3 MEDIUM + 5 small??? Why 3 MEDIUM? It is worth adding at least two graphic examples here. Equation (1) is incomprehensible.

4.       Caption under Figure 6 - instead of: small (a), medium (b), and large (c)., it should be: (a) small; (b) medium; (c) large. Check the Instructions for Authors.

5.       In lines 425–431, according to the Instructions for Authors, there should be initials. Check the Instructions for Authors.

Author Response

Dear Reviewer 2,

Please find attached a cover letter that addresses the second round of minor revisions. You will find the revisions in blue text.

We very much appreciate your role in helping to improve our paper.
